# Acute Onset Quadriplegia and Stroke: Look at the Brainstem, Look at the Midline

**DOI:** 10.3390/jcm11237205

**Published:** 2022-12-04

**Authors:** Marialuisa Zedde, Ilaria Grisendi, Francesca Romana Pezzella, Manuela Napoli, Claudio Moratti, Franco Valzania, Rosario Pascarella

**Affiliations:** 1Neurology Unit, Neuromotor and Rehabilitation Department, Azienda USL-IRCCS di Reggio Emilia, 42122 Reggio Emilia, Italy; 2Stroke Unit, Dipartimento di Neuroscienze, AO San Camillo Forlanini, 00152 Roma, Italy; 3Neuroradiology Unit, Radiology Department, Azienda USL-IRCCS di Reggio Emilia, 42122 Reggio Emilia, Italy

**Keywords:** cerebrovascular disease, medial medullary infarction, heart appearance sign, stroke, brain MRI, quadriplegia, vertebral artery

## Abstract

Acute onset quadriplegia with or without facial sparing is an extremely rare vascular syndrome, and the main focus of attention is on the cervical and upper thoracic spinal cord as the putative site of the damage. Quadriplegia has been occasionally reported in brainstem strokes within well-defined lesion patterns, but these reports have gained little attention so far because of the rarity of this clinical syndrome. The clinical, neuroanatomical and neuroimaging features of ischemic stroke locations associated with quadriplegia have been collected and reviewed in a pragmatical view, which includes a detailed description of the neurological signs associated with the damage of the pyramidal pathways. Two clinical examples have been added to raise practical suggestions in neurovascular practice. Ischemic stroke sites determining quadriplegia have some main well-defined midline locations in the brainstem, involving the pyramidal pathways of both sides in a single synchronous ischemic lesion in the medulla oblongata and in the pons. Several accompanying neurological signs have been described when the ischemic lesion involves tracts and nuclei other than the pyramidal pathways, and they can be useful as localizing clues. In some cases, the typical neuroimaging appearance of the ischemic lesion on Magnetic Resonance Imaging (MRI) has been reported as being a “heart appearance sign”. This last sign has been described in midbrain strokes too, but this location is not associated with quadriplegia. The main etiology is atherothrombosis involving the intradural segment of the vertebral artery (VA) and their perforating branches. Two clinical examples of these rare vascular syndromes have been chosen to support a pragmatical discussion about the management of these cases. A midline ischemic stroke in the brainstem is a very rare vascular syndrome, and the acute onset quadriplegia is a distinctive feature of it. The awareness of this cerebrovascular manifestation might help to recognize and treat these patients.

## 1. Background

Quadriplegia or tetraplegia is the term used to refer to total weakness of all four limbs which may be of an upper motor neuron type or less commonly, a lower motor neuron type. The anatomic localization of acute atraumatic quadriplegia is challenging, and a broad differential diagnosis includes disorders affecting the upper motor neurons in the brainstem or more commonly, the cervical spinal cord, as well as disorders of the lower motor neurons, including acute motor neuropathy or neuromuscular junction disease. Quadriplegia has been described in spinal cord injuries with a wide range of causes, including spinal cord ischemia [1,2]. Spinal cord ischemia involving the anterior horns and tracts of the cervical and upper thoracic levels is characterized by the association of lesional level signs of lower motor neurons and signs of the upper motor neurons’ deficit under the lesion level. These last ones are related to the bilateral damage of the pyramidal pathways in the anterolateral tracts of the spinal cord, which are supplied by the anterior spinal artery (ASA). One of the main clinical hallmarks of a vascular cause of spinal cord injury is the abrupt-to-subacute onset of it and the progression of hyposthenia within few hours (12 h according to the recently proposed diagnostic criteria) [3]. Within this early time, interval neuroimaging investigations usually are not able to directly identify a spinal ischemia, and Diffusion-Weighted Imaging (DWI) MRI sequences in the early phase (hours from symptom onset) are usually hard to interpret, and an hyperintense signal change in the spinal cord (owl eyes and pencil-like pattern) may be not yet visible on T2-weighted sequences [3,4,5,6,7]. The initial MRI of the spinal cord may be normal in up to 24% of the patients [3]. 

Moreover, in rare cases, the delayed MRI of the spinal cord may not detect signs of spinal cord involvement despite there being persisting quadriplegia, and sometimes, other neurological signs may develop including cranial nerves signs and gaze limitations, pointing to a brainstem location of the damage. There are few and extremely rare vascular syndromes of the brainstem with a bilateral paramedian lesion of the pyramidal pathways which are known to be responsible for an acute onset of quadriplegia.

In this narrative review the main locations and clinical and neuroimaging features of these syndromes are addressed.

## 2. Neuroanatomical Issues

### 2.1. Corticospinal Tract in the Brainstem

The corticospinal tract arises from the precentral gyrus (somatomotor area, area 4) and from the postcentral gyrus (areas 3, 1 and 2), ending primarily in laminae VI–IX and in laminae IV and V of the anterior horns of the spinal cord, respectively. Frontal area 6, and parietal areas 5 and 7 also contribute to the corticospinal tract. The lateral corticospinal tract is the most clinically relevant descending motor pathway in the nervous system, and a lesion in any point (step) of its course will produce a typical strength deficit that often enables the precise clinical localization of it. The primary motor cortex neurons contributing to the corticospinal tract are located mostly in cortical layer 5, and they are named pyramidal cells (about 3% of corticospinal neurons are giant pyramidal cells called Betz cells). The axons of the pyramidal cells from the cerebral cortex enter the upper portions of the cerebral white matter (corona radiata) and descend toward the posterior limb of the internal capsule. The lateral corticospinal tract contains also fibers projecting from the cortex to the brainstem, including the motor fibers of the face (corticobulbar fibers). The internal capsule continues into the midbrain cerebral peduncles in the ventral portion (basis pedunculi). The middle one-third of the basis pedunculi contains corticobulbar and corticospinal fibers with the face, arm, and leg axons being arranged from the medial to the lateral directions, respectively. The other portions of the basis pedunculi contain primarily corticopontine fibers. Next, the corticospinal tract fibers descend through the ventral pons, where they form somewhat scattered fascicles, which collect on the ventral surface of the medulla to form the medullary pyramids. In truth, the pyramids include reticulospinal and other brainstem pathways in addition to the corticospinal tract. The transition from the medulla to the spinal cord is called the cervico-medullary junction, which occurs at the level of the foramen magnum. At this point about 85% of the pyramidal tract fibers cross over in the pyramidal decussation to enter the lateral white matter columns of the spinal cord, forming the lateral corticospinal tract. The remaining ~15% of the corticospinal fibers continue into the spinal cord ipsilaterally, without crossing, and they enter the anterior white matter columns to form the anterior corticospinal tract [8,9].

Bilateral lesions of the pyramidal tracts in the medulla and pons may cause quadriplegia, so this review is focused on the vascular supply and lesional pattern of these portions of the brainstem. Instead, midbrain paramedian bilateral lesions do not cause quadriplegia because the cortico-spinal tracts are in the anterolateral region, so they have been only marginally mentioned in this review.

### 2.2. General Vascular Organization and Territories of the Brainstem

The main arterial supply of the brainstem [10,11,12] is provided by the vertebro-basilar system, including the vertebral artery (VA), basilar artery (BA), anterior and posterior spinal arteries ASA and PSA), posterior inferior cerebellar artery (PICA), anterior inferior cerebellar artery (AICA), superior cerebellar artery (SCA), posterior cerebral artery (PCA) and anterior choroidal artery (AChA). The collaterals of these arteries are divided into four arterial groups (anteromedial, anterolateral, lateral and posterior) according to their point of penetration into the parenchyma. Each one of these groups supplies the corresponding arterial territories in the brainstem. The arterial territories have a variable extension at different levels of the brainstem.

The medulla oblongata is supplied by the VAs and the PICAs, which give rise to the rami of the lateral medullary fossa, and by the ASA and PSAs.

Different arterial trunks supply blood to the pons including the VA, the AICA, from which the rami of the lateral medullary fossa arise, the SCA and the BA.

Five arterial trunks supply the midbrain: the SCA (mainly the medial branch), the collicular artery, the posteromedial choroidal artery, the middle rami of the interpeduncular arteries arising from the PCA and the AChA arising from the carotid system.

In more detail, the vascular zone of the medulla is divided into four territories: the large anteromedial, small anterolateral, large lateral and small dorsal ones. The anterolateral and posterior zones are smaller than the others. The arterial supply of each territory is summarized in Table 1.

At the pontine level, three intrinsic arterial territories can be identified: the anteromedial, anterolateral and lateral territories. The anteromedial pontine territory is supplied by distinct arterial sources arising from different levels of the BA (foramen coecum arteries, pontine arteries and inferior rami arising from the interpeduncular fossa arteries). The posterior territory only exists in the upper part of the pons. The arterial supply of each territory in the pons is summarized in Table 2. 

Each vascular zone includes different cranial nerve nuclei and fascicles. Infarcts in different vascular zones may cause various symptoms [10,12], so the clinical picture of medullary and pontine infarcts is variable. The identification of the affected vascular territory in a medullary stroke helps to determine the etiology of it.

The midbrain’s vascular supply is quite elaborate, and it depends on the midbrain level, where the contribution of the PCA increases caudo-rostrally. Four vascular territories are relatively constant; the most often ones that are involved are the anteromedial (paramedian) and anterolateral territories at the middle and upper midbrain, which are followed by the lateral and dorsal territories [12]. An isolated midbrain bilateral stroke does not produce quadriplegia, so the discussion of the clinical, topographical and vascular issues of the rarest vascular syndromes of the midbrain inconstantly associated with quadriparesis and the anatomical reasons as to why quadriplegia is not among the neurological signs of midbrain stroke are described in Section 3.3.

A graphical distribution of vascular territories of the brainstem is illustrated in Figure 1.

## 3. Brainstem Midline Stroke

Most medullary and pontine strokes are sharply marginated and paramedian, with the long axis being oriented in the sagittal plane. This pattern is related to the distribution of the paramedian-penetrating branches arising from the BA and distal VAs, which perforate the paramedian brainstem and never cross the midline. Lateral infarcts, in the distribution of the short circumferential arteries, are seen less frequently than paramedian lesions are. At the midbrain level, midline infarctions can be visualized, as the many involved branches are not limited to a paramedian distribution.

Characteristically, brainstem vascular syndromes are also called alternating syndromes due to the presence of crossed neurological signs (ipsilateral cranial nerve signs and contralateral signs of the ascending and descending tracts) which are hallmarks of the brainstem location. However, some subtypes of medullary and pontine syndromes are not in this scheme: e.g., the clinical triad of unilateral medial medullary syndrome usually does not have crossed signs, and in general, the patients with medullary strokes may also present a broad range of untypical, life-threatening and hard-to-recognize features [12,13,14]. 

### 3.1. Medullary Infarction

The medulla oblongata can be classified into three major portions, which are summarized in Table 3.

The upper part of the posterior surface of the medulla oblongata also forms the lower floor of the fourth ventricle which contains the area, postrema.

The caudal regions of the medial medulla oblongata are supplied by paramedian branches of the ASA (which arises from both the VAs), whereas more rostrally located regions of the medial medulla oblongata are supplied by the paramedian branches of the VAs [10]. The lateral medulla oblongata is mostly supplied by the circumferentially penetrating branches from the VA, while the PICA supplies the remaining lateral and posterior portion of the medulla oblongata.

The vascular territories, anatomical segments and supplying arteries are illustrated in Figure 2.

Two main vascular medullary syndromes can be distinguished: the lateral medullary syndrome (Wallenberg’s syndrome), which is the most common one (about 2% of ischemic strokes), and the medial medullary syndrome (Dejerine’s syndrome), which has an incidence up to four times lower than that of the lateral medullary syndrome (less than 1% of all brainstem strokes) [15]. At least three other less frequent and overlapping syndromes may occur by the spreading of an ischemic lesion and occlusions of several VA branches.

Medullary infarctions account for 7% of all ischemic brainstem strokes [16]. The most frequent cause of medial medullary infarctions (MMI) is the atherosclerosis of the VA and its branches (mainly ASA), but VA dissection too has been reported in a selected population as a cause. 

Unilateral MMIs produce Dejerine’s Syndrome, whose clinical features are represented by the classical triad of contralateral hemiparesis/hemiplegia, contralateral loss of position and vibration sense and ipsilateral tongue weakness; in addition, oculomotor abnormalities and dysarthria may be present. The involved structures and corresponding deficits in MMI are summarized in Table 4. 

The most common presentation of unilateral MMI in a case series was hemiparesis; the second most common clinical manifestation was hemisensory symptoms, which in a few patients involved the face [17]. Tongue weakness was an uncommon finding, and it was less frequent than facial weakness. 

Since the pyramidal tract, medial lemniscus and hypoglossal nucleus are arranged anterior-posteriorly in the medial medulla, the clinical presentation is dependent upon the anteroposterior extent of the lesion; a lesion involving the anterior medulla sparing the posterior aspect will result in the sparing of the hypoglossal nucleus, thereby explaining the absence of tongue weakness. Nystagmus was another finding which was seen in a few patients. 

Bilateral MMI (BMMI) is a rare event, and basically, the features summarized in Table 4 for the unilateral MMI are present on both sides, with the characteristics of quadriplegia and gaze paralysis, which may be confused with some features of locked-in syndrome. If quadriplegia/quadriparesis is a characteristic clinical hallmark, an upper motor neuron type of facial weakness may be unilateral, and the sensory manifestations or nystagmus are not constant. The facial palsy in patients with a medullary infarction has been postulated to result from the interruption of aberrant cortico-facial fibers which descend to the level of the upper-middle medulla and are located ventromedially, decussate, and then ascend in the dorsolateral medulla to supply the facial nucleus [18] (Figure 3). 

In a systematic review of 38 cases of MRI-proven BMMIs [20] described from 1992 to 2011, the mean age was 62.2 years, and 74.2% of them were male. The most common clinical presentations were motor weakness in 78.4% of them, dysarthria in 48.6% of them, and hypoglossal palsy in 40.5% of them as rostral medullary lesions. Thirty-eight point-five percent of the patients had VA atherosclerosis as a putative etiologic mechanism, which was followed by branch occlusive disease. A characteristic, although uncommon, is the neuroimaging sign in brain MRI which has been described as having a “heart appearance sign” [21,22,23]. A good example of the MRI of the “heart appearance sign” both on Diffusion Weighted Imaging (DWI) and in Fluid Attenuated Inversion recovery (FLAIR) sequences is showed in Figure 4. 

The patient whose MRI was shown in Figure 4 was admitted to the ED because he had a transient right hemisensory syndrome and he developed during the hospital stay, the abrupt onset of hypotonic quadriplegia, severe dysarthria and respiratory failure, and he required ventilator support. In the following hours, bilateral hypoglossal palsy, complex gaze abnormalities with horizontal gaze limitation and profound anesthesia for touch and pain in the four limbs and trunk developed. The cause of the BMMI was an atheromatous right V4 VA occlusion (Figure 5).

The treatment strategy in the acute phase was a double antiplatelets regimen with high dosage of statins associated with low-molecular weight heparin (LMWH) for the prevention of venous thromboembolism. 

Another even rarer subtype of medullary infarction is the infarction of the pyramidal decussation, which is scarcely described in the literature [24,25,26,27,28]. The expected clinical syndrome involves quadriplegia with the sparing of the face and sensory fibers. The proposed mechanism is the occlusion of the perforators off of the ASA, because at the level of the pyramidal decussation, the ASA does not supply the sensory tracts, which are instead usually fed by the VA and PSAs, thus explaining the absence of sensory deficits. Moreover, anarthria and dysphagia might be a sign of the involvement of the hypoglossal nucleus and fibers, with perhaps some involvement of the cranial nerves IX and X. 

### 3.2. Pontine Infarction

Pontine infarctions are relatively rare, accounting for only about 7% of all ischemic strokes of the brain [29,30,31,32]. Considering the brainstem strokes, the pons, either in isolation or as part of multifocal ischemia, is involved much more often than the other brainstem structures are. Isolated pontine infarcts account for 12–27% of posterior circulation ischemia [15,33].

Based on clinical, anatomical and neuroradiological correlations, the following constant vascular territories have been defined in pontine ischemia: ventromedial (anteromedial), anterolateral, lateral (tegmental), dorsal and bilateral infarcts (Figure 1b). The relative size and extension of each vascular territory may show a variability both in the causal, middle and rostral pons and in the axial distribution.

On the longitudinal axis, the infarction can occur in the caudal (lower) pons, middle or rostral (upper) pons or in two or three of these portions. In the axial plane, the paramedian, anterolateral, lateral or posterior region and their various combinations may be affected [29,30,31,32,33,34,35,36,37,38,39,40,41,42,43].

Ventral (paramedian) infarcts are the most common location (two out of three isolated pontine infarctions), and they are caused by the occlusion of the anteromedial or anterolateral-perforating arteries coming off from the BA through the BA plaques; conversely, small deep pontine infarctions are more frequently to be associated with small vessel disease neuroimaging signs [34,35].

Perforating arteries are on average, unilaterally, 5.8 mm in size (range: 4–10), and they have a mean diameter of 0.39 mm (range: 0.09–0.81 mm) [35]. They can be divided into the caudal group, entering the foramen caecum, the middle group, penetrating the edges of the basilar groove, and the rostral group, which entered the most caudal part of the interpeduncular fossa (see also Section 3.3). Some of them always arise from the BA, either separately or by their own common trunks, or by common stems with some of the BA leptomeningeal vessels. Sixty-two point-five percent of these pontine-perforating arteries give off 1–3 anterolateral twigs [35], and a branch coming off unilaterally may supply both the ipsilateral and contralateral paramedian region of the pons. Similarly, the caudal pontine branches may nourish the upper part of the medullary pyramid and the olive. The BA gives origin to the leptomeningeal branches too, including the short pontomedullary (PMA) and anterolateral arteries (ALAs), the long inferolateral pontine artery (ILPA), the superolateral (SLPA) and posterolateral ones (PLPA), as well as some branches of the cerebellar arteries (AICA, SCA and more rarely PICA). The short ALAs have multiple (mean, 5.2) vessels on each side, arising directly from the BA or from the perforating arteries and less frequently from the long leptomeningeal or cerebellar arteries [35,36]. The PMA is a short BA branch which ends superiorly to the medullary olive. The ILPA, a single vessel, runs inferiorly and laterally, and it ends just below the level of the trigeminal nerve root, but it can give rise to the ALAs, rarely to a single perforating artery, and always to the lateral intrapontine branches.

The SLPA usually gives off one or two ALAs, rarely a perforating artery and very often lateral twigs, ending at the level of the trigeminal nerve root. The PLPA arises just below the origin site of the SCA, and it ends above the trigeminal nerve root.

The cerebellar arteries may contribute to the arterial supply of the pons: the AICA mainly gives off twigs to the lateral part of the lower pons, and the SCA to the most posterior (dorsal) part of the rostral tegmentum. Some perforating arteries and ALAs also can arise from the cerebellar arteries.

Several anastomoses have been described both between the neighboring perforating branches (25% of the cases) and among the SLPA and ILPA branches (56.3% of the cases) [35]. 

The medial basis pontis (corticospinal tract) is supplied by the short midline perforators branching directly off the BA, from which the long midline perforators supply the medial tegmentum (including the medial part of the medial lemniscus, the abducens nucleus, medial longitudinal fasciculus and paramedian pontine reticular formation).

Paramedian infarcts often extend to the ventral surface, whereas an ischemia restricted to the paramedian tegmentum is less common [33]. The clinical hallmark of a ventral pontine infarction is contralateral hemiparesis with a moderate-to-severe severity, and it is more marked in the upper extremity and in the distal part of the limbs. It may be an isolated neurological deficit such as pure motor hemiparesis in about one half of the ventral pontine infarctions [33]. Distinct lingual, contralateral palatal–lingual or palatal–lingual–laryngeal hemiparesis are uncommon patterns of motor deficit and resemble those observed in capsular genu syndrome. Moderate or marked dysarthria is almost a constant finding in large paramedian infarcts, particularly of the upper pons, and it is often accompanied by hemiparesis, brachial monoparesis, supranuclear facial palsy and hemiataxia, while an isolated dysarthria is a rare occurrence [37].

The unilateral anteromedial infarcts cause contralateral hemiparesis/hemiplegia, contralateral ataxia, dysarthria, dysphagia, nystagmus, and often, ipsilateral facial palsy. The associated less frequent signs are the contralateral loss of proprioception, paresis of the ipsilateral horizontal gaze and internuclear ophthalmoplegia (INO). The involved structures and corresponding deficits in anteromedial pontine infarcts are summarized in Table 5.

The paramedian pontine infarctions may show complete (anteriomedian and anterolateral territories), incomplete and partial paramedian lesions, accounting for different neurological deficit from the ones illustrated in Table 5. The complete paramedian ischemia extends along the entire raphe pontis and damages the pyramidal, corticobulbar, pontocerebellar and tectospinal tracts, the medial lemniscus and the medial longitudinal fasciculus (MLF), as well as the abducent nucleus, the genu of the facial nerve and the paramedian pontine reticular formation (PPRF) in the lower pons. A paramedian ischemia in the middle pons, where the pyramidal bundles are not compact, can cause hemiparesis with arm predominance or brachial monoparesis due to a lesion of the ventromedial pyramidal fascicles for the hand and arm [37]. 

Bilateral anteromedial pontine infarctions are a rare entity, which is similar to BMMI, and also in these cases, a “heart appearance sign” has been described [33,38]. In a single case [39], it was postulated that a pontine infarct may assume the shape of a heart if the arteries supplying the bilateral anteromedial and anterolateral territories are involved. In Figure 6, there is an image of a schematical view of the pontine structures involved in the heart and the corresponding arterial supply.

In Figure 7, a neuroimaging example of an incomplete “heart appearance sign” at the pontine level is depicted. The brain MRI was performed within 12 h from the symptom onset (quadriplegia, bilateral facial weakness, severe dysarthria and dysphagia). 

### 3.3. Midbrain Infarction

Midbrain infarcts account for 2% of all cerebral infarcts, whereas their frequency increases up to 8% in the infarcts limited to the posterior circulation [40,41,42]. Isolated midbrain infarcts are infrequent because of the midbrain’s common vascular supply with other infra- or supratentorial anatomical sites, and the incidence of isolated midbrain infarction varies only from 0.7% to 2.3% [42]. Midbrain infarcts more often coexist with infarcts in neighboring structures (diencephalon and pons) or the temporo-occipital cortex and superior cerebellum [40,42]. 

Bilateral midbrain infarcts are an extremely rare vascular syndrome, and they are located in the anteromedial (paramedian) vascular territory, which is supplied by the penetrating perforators from the BA, SCA and P1 PCA. The paramedian territories of the midbrain and thalamus are supplied by interpeduncular perforating branches. In previous anatomical studies, the interpeduncular branches have been described as three groups of vessels that originate from the last 5 mm of the basilar artery from the initial 7 mm of both the SCAs and from the P1 PCA [43] (Table 6). 

SPMA and PThA frequently originate as a single or ‘common’ trunk, known as the type IIb variant of the artery of Percheron [44]. The interpeduncular perforating branches have great variability with respect to their number, size, origin and territorial contribution to the midbrain and thalamus [43]. This can result in a variable range of imaging appearances in stroke patients when the vascular disease involves the interpeduncular perforating branches. It is easily inferable from the anatomical organization summarized in Table 6 that only the IPMA territory is associated with an isolated midbrain infarction, and the IPMA involvement produces a caudal paramedian midbrain infarction (CPMI), and this perforating artery may origin from the distal BA and proximal PCA or SCA. 

The anteromedial segment of the midbrain contains critical structures for the vertical gaze (posterior commissure, periaqueductal region and the rostral interstitial nucleus of the medial longitudinal fasciculus) at the upper midbrain tegmentum. Therefore, infarcts involving the territories of the paramedian perforators of the BA, perforating branches of the SCA or the posterior thalamo-subthalamic paramedian artery (branch of the P1 PCA) usually produce vertical gaze palsies. Bilateral upper midbrain infarcts are characterized by a wide range of conjugate or disconjugate supranuclear vertical gaze palsies (rarely in isolation) [40,41,42,45]. A complete bilateral ophthalmoplegia (bilateral ptosis with loss of all extraocular movements) is an unusual sign of bilateral infarcts at the meso-diencephalic junction. Midbrain paramedian rostral infarcts can also manifest without gaze palsies, including pure motor hemiparesis or isolated gait ataxia, but they have been described only in unilateral lesions. When the ischemia is located around the red nucleus, the predominant feature may be body lateropulsion (contraversive falls to the side of lesion) due to interruption of ascending fibers of the crossed dentate-rubro-thalamic pathway.

Nuclear third nerve palsy (isolated or associated with hemiparesis or ataxia) is a localizing sign of the paramedian territory infarct at the middle level of the midbrain. The most common pattern of the nuclear oculomotor disorder is ipsilateral third nerve palsy with contralateral superior rectus paresis, which often presents with bilateral ptosis and mydriasis. Dysarthria, which is often associated with other signs, can be found in half of the patients with a pure midbrain infarction, whereas dysarthria (hypokinetic and palilalia) as the prominent clinical manifestation can be due to a single ischemic lesion involving the medial ventral part of the substantia nigra. Acquired stuttering can, in rare instances, be the outstanding manifestation of a small midbrain ischemia. 

At the lower paramedian midbrain level, tetraataxia—which is often associated with hemiparesis, delayed tremor or palatal myoclonus—is a rare occurrence following bilateral, or less commonly, unilateral infarcts involving the crossing efferent dentato-rubral fibers [40]. Uni- or bilateral internuclear ophthalmoplegia (INO) due to the involvement of the medial longitudinal fasciculus together with limb or gait ataxia, dysarthria and tremor can reveal a caudal paramedian infarct, whereas an isolated bilateral INO is less common. Pathological laughter, although very rare, can herald a paramedian lower midbrain infarct (associated with dysarthria and hemiparesis) that disrupts the brainstem basal ganglia–forebrain circuitry which participates in laughter control. The caudal paramedian midbrain includes small, extremely medial portions of the cerebral peduncles (fronto-pontine tract), extremely medial portions of the substantia nigra, the superior cerebellar peduncles and their decussation, the central tegmental tract (the component of Mollaret’s triangle) and its decussation, the medial longitudinal fasciculi (MLF), the nuclei of the trochlear nerve (CN 4), the short segment of the CN4 fibers and the reticular structure [10]. Anatomically, the decussation of the superior cerebellar peduncles and the central tegmental tract constitute the horseshoe-shaped Wernekink’s commissure that is named after German anatomist Friedrich Wernekink [46]. A caudal paramedian midbrain infarction (CMPI) is an extremely rare form of an ischemic stroke (representing 12 out of 6820 cerebral infarction patients in the biggest published series) [47], and bilateral infarcts are even rarer (5/12 of them in the same case series). Neuroimaging features are a backward oblique sign in the lower level of the midbrain in unilateral sagittal MRI images, and they have a characteristic “V-shaped” appearance in the axial MRI in bilateral infarcts. In all of the patients, the neurological signs are bilateral cerebellar dysfunction (dysarthric speech, truncal or gait ataxia and four-limb ataxia), diplopia and INO, but not hemiparesis. CPMI is a rare cerebrovascular disease that destroys the Wernekink’s commissure, medial longitudinal fasciculi and other adjacent structures, but it usually does not involve the corticospinal tracts. The primary mechanisms of unilateral CPMI involve small vessel disease. The underlying stroke mechanisms of bilateral CPMI are either large artery atherosclerosis disease or a cardiac embolism. In bilateral CPMI cases, a “heart sign” has been reported in diffusion-weighted imaging (DWI) and fluid attenuated inversion recovery (FLAIR) images [48], which is smaller than the corresponding hearts in the medulla and pons. The clinical pattern of the infarct corresponds to the above described Wernekink’s syndrome (bilateral cerebellar dysfunction, wall-eyed bilateral internuclear ophthalmoplegia or WEBINO) without delayed palatal myoclonus. The Wernekink’s commissure consists of crossed dentate-rubro-thalamic tracts and dento-rubro-olivary bundles [46]. Bilateral cerebellar dysfunction is attributed to the interruption of the dentate-rubro-thalamic pathways prior to and after decussation. WEBINO is ascribed to the destruction of the medial longitudinal fasciculus (MLF) near the Wernekink’s commissure. Delayed palatal myoclonus and hypertrophy olivary degeneration in the medulla oblongata caused by primary lesions in the dento-rubro-olivary pathway have been reported in a few cases.

The “heart appearance” sign in the medulla oblongata or pons was considered to appear when the infarct occurred in the anterior-medial and anterior-lateral territories [49]. However, for a “heart appearance” in the caudal midbrain infarction, resulting in Wernekink’s commissure syndrome, it should involve the medial and lateral paramedian branches on both sides, sometimes with a stepwise timing. 

Hemiparesis is not the typical feature of midbrain paramedian infarcts, so even bilateral paramedian infarcts are not associated with double hemiparesis and even less so to quadriplegia. Bilateral infarcts are uncommon, occurring mainly when both of the paramedian territories at the upper level are affected. In such cases, the ischemia may extend to the neighboring thalamus. Different patterns of vertical gaze palsy are the main clinical findings. Locked-in syndrome has been found to be associated with restricted ischemia, affecting both the cerebral peduncles. Oculomotor signs predominate along with hemiparesis and hemiataxia when combined anteromedial and anterolateral infarcts occur [40,42]. In the case series of forty patients with Asian ancestry described by Kim et al. [42], pure midbrain anteromedian infarcts are not associated with hemiparesis, but the anterolateral infarctions showed definite hemiparesis in 30% of the cases. A similar clinical picture has been provided by a case series in a different population sample [41].

A more complex and anecdotal vascular syndrome of the midbrain, which is associated with bilateral pyramidal tract damage, is the bilateral cerebral peduncular infarction (BCPI) [50,51,52]. Data from one medical center show that BCPI accounts for 0.26% of all of the admitted patients with an ischemic stroke [53], but the main limitation is that they were not pure midbrain infractions, and the other sites of infarction were found in the thalamus, pons and cerebellum. Moreover, the case described by Asakawa et al. [54] proposing the Mickey Mouse ears sign as a feature of this infarct the midbrain involvement was accompanied by a cerebellar infarction. Interestingly, two reported cases [50,51] had normal muscular strength. The most common symptoms of isolated BCPI included ataxia, dysarthria, sensory disturbance and mild paresis of the extremities, with a few disorders of eye movement or light reflex appearing upon a neurologic examination (rare involvement of the oculomotor nerve if the paramedian area is interested). The nerve fibers of the corticospinal tract (CST) run in the pedunculus cerebri, reaching the cerebellum via the cortico-ponto-cerebellar tract (CPCT). The damage to the CST and CPCT in the pedunculus cerebri is the cause of the mild paresis of the extremities, dysarthria and ataxia in BCPI. The reason why only mild hemiparesis was present even when the pyramidal tract at the crus cerebri was heavily involved has not been convincingly explained. One possible explanation for this is that the patients with hemiparesis also consistently showed the representation of the lower limb in the lateral part of cerebral peduncle, with the face and upper limb being represented more medially [40]. 

The vascularization of the midbrain is complex because there is a significant contribution by the perforating branches of the posterior communicating arteries and the peduncular perforating arteries and circumflex branches of the P1 or P2 segment of the PCA in addition to the supply through the BA and cerebellar arteries. The blood supply on the cerebral peduncle is achieved through multiple branches of the PCA, the SCA or the interpeduncular perforating branches originating from the tip of the BA. These arteries are known as the medial mesencephalic branches (MMB). [55]. Therefore, the blood supply on the cerebral peduncle mostly comes from the PCA and SCA, which are located at the distal segments of the BA. The perforating branches of the P1 PCA may play an important role in isolated cerebral peduncular infarction [53]. 

## 4. Clinical Hints and Differential Diagnosis

Acute atraumatic quadriplegia in the absence of spinal cord lesions represents a rare clinical entity, and it is globally difficult to deal with on the cerebrovascular side. The rarity of non-spinal ischemic cases causes makes constructing a hypothesis about this diagnosis difficult in many contexts. There are some aspects that can help in selecting the correct diagnostic course to establish a suspected cerebrovascular disease (summarized in Table 7).

Firstly, the onset is almost always acute, which is often hyperacute, and the strength deficit, which is often homogeneous to the four limbs, reaches its maximum severity within a few seconds or minutes, and much more rarely, hours, which is unlike what happens in spinal ischemia.

Secondly, it is possible that the previous clinical history shows the occurrence of transient neurological deficits that points to the affected territory (e.g., pathological laughing in pontine stroke) or that in any case, even if not specific (e.g., hemisensory loss or hemiparesis), it points towards a cerebrovascular cause.

Thirdly, in some sites of the lesion of the brainstem, it is possible that there is a facial saving, but this element is not constant, as evidenced in the described cases of BMMI and pontine paramedian infarctions, due to the involvement of the nucleus and fibers of the VII cranial nerve. Only the very rare infarction of the decussation of the bulbar pyramids is clearly associated with a facial sparing. Therefore, an infarction of the pyramidal decussation must be considered in patients with pure motor quadriplegia.

Fourthly, even when quadriplegia/quadriparesis is the primary neurological deficit, it is possible to identify additional neurological signs that aid in localizing the injury site, such as abnormal oculomotion or cerebellar signs or signs of involvement of individual cranial nerves (such as the XII cranial nerve in the BMMI).

Fifthly, a lesion evolution has frequently been described, which occurs in the first hours or days after the onset of symptoms and which corresponds to the involvement of the contiguous vascular territories (e.g., in addition to the anterolateral to the anteromedial territory in pontine paramedian infarcts or posterior extension in BMMIs such as gradually involving the lemniscal structures, the medial longitudinal fascicle and the nucleus of the hypoglossal nerve).

Most of the patients described in the literature are men from the sixth to the seventh decade of their lives and with a composite vascular risk profile. Moreover, this issue can be of help in orienting the diagnostic suspicion, although theoretically, it does not allow us to exclude other causes.

The consideration of suspicions different from the cerebrovascular one, i.e., of a peripheral type (e.g., acute motor axonal neuropathy—AMAN; variant of Guillain-Barré syndrome) or of an infectious type (encephalitis of the brainstem), is possible, and it has been reported in the literature but in a different clinical context and sometimes with diagnostic delay in relation to cerebrovascular genesis.

## 5. Conclusions

The assessment and management of acute nontraumatic quadriplegia without spinal cord lesions is a challenge, and it needs a precise knowledge of the vascular anatomy and territories of the brainstem. Even so, these are very rare events, but they have a characteristic clinical presentation pattern that is often evocative of the site of the damage. The consideration of these clinical entities helps us to suspect them in the face of the individual case, in which diagnostic and therapeutic decisions must be made quickly as this is a time-dependent condition. 

## Figures and Tables

**Figure 1 jcm-11-07205-f001:**
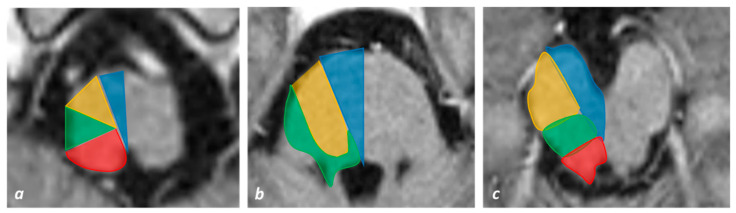
**Vascular territories of the brainstem superimposed on axial T1W MRI slices**. (**a**) Vascular territories at the level of the medulla oblongata: anteromedial group arteries (blue), anterolateral group arteries (yellow), lateral group arteries (green) and posterior group (red). (**b**) Vascular map at the level of the pons: anteromedial group (blue), anterolateral group (yellow) and lateral group (green). (**c**) Vascular territories at the level of the midbrain: anteromedial group arteries (blue), anterolateral group arteries (yellow), lateral group arteries (green) and posterior/dorsal group (red).

**Figure 2 jcm-11-07205-f002:**
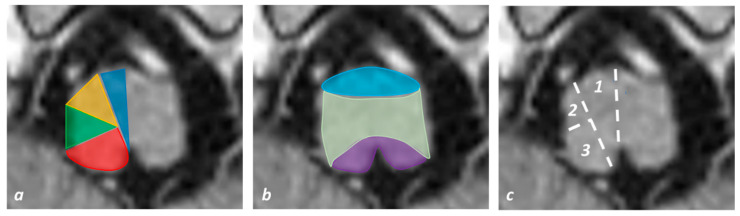
**Vascular territories, anatomical division and main supplying arteries of the medulla oblongata superimposed on an axial T1W MRI slice**. (**a**) Vascular territories at the level of the medulla oblongata as in Figure 1a: anteromedial group arteries (blue), anterolateral group arteries (yellow), lateral group arteries (green) and posterior group (red). (**b**) Anatomical segments as in Table 3: anterior segment (light blue), tegmentum (light green) and posterior segment (violet). (**c**) Main supplying arteries: VA (paramedian branches) and ASA (1); VA (2); VA and PICA (3).

**Figure 3 jcm-11-07205-f003:**
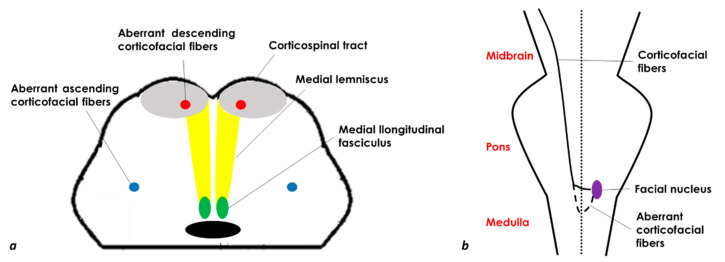
**Schematic drawing of a medulla axial section (a) and a coronal section of the brainstem with the aberrant corticofacial fibers (b) (modified and redesigned from [19])**.

**Figure 4 jcm-11-07205-f004:**
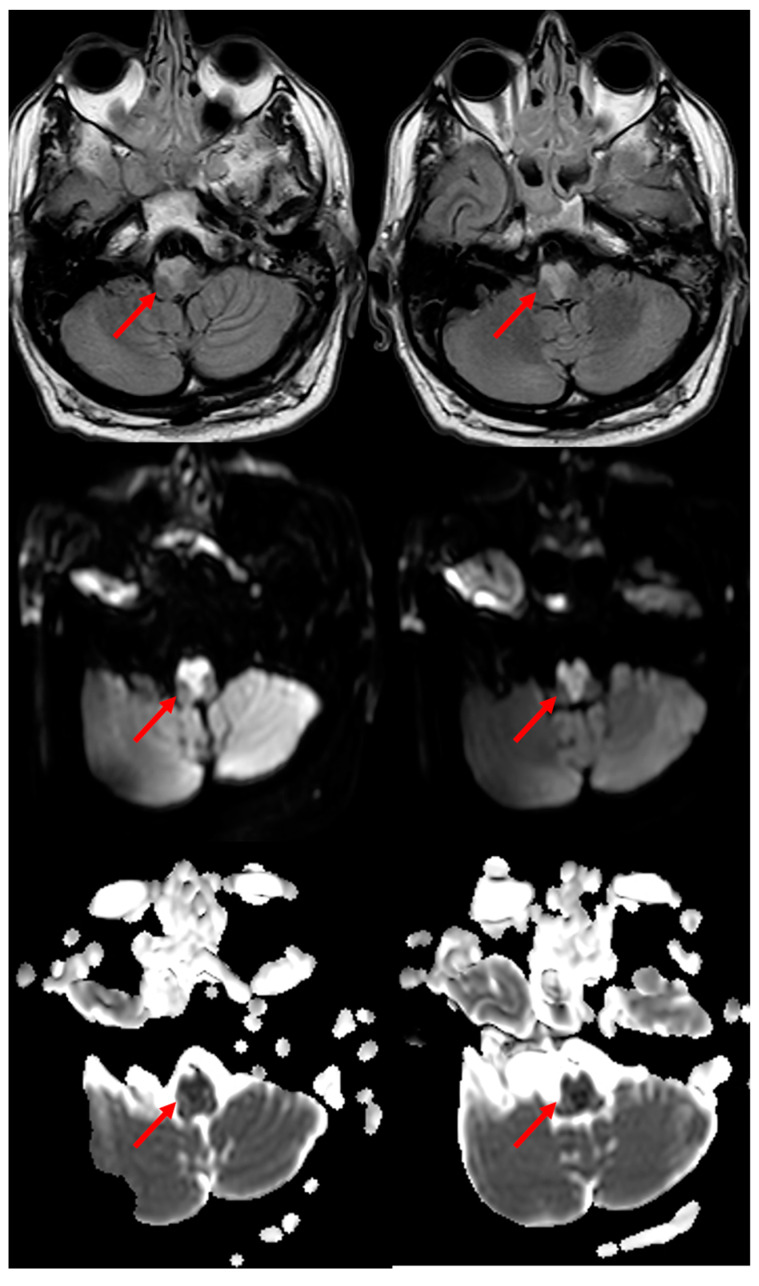
**“Heart appearance sign” in acute BMMI, which is well evident in MRI on FLAIR axial images (first row), DWI images (second row) and Apparent Diffusion Coefficient (ADC) images (third row) (red arrows)**.

**Figure 5 jcm-11-07205-f005:**
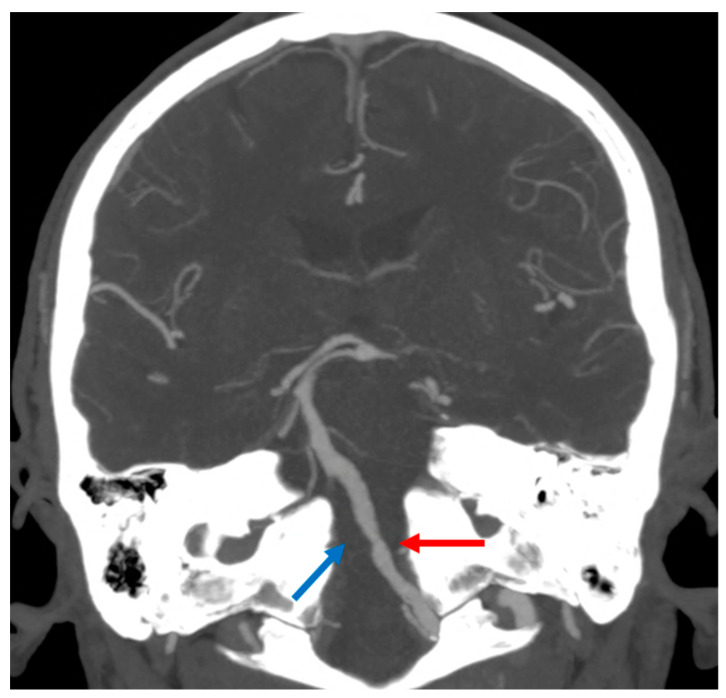
**Intracranial Computed Tomography Angiography (CTA) with Maximum Intensity Projections (MIP) reconstruction in coronal plane, showing atheromatous changes in left V4 VA (red arrow) and occlusion of the right V4 VA (blue arrow)**.

**Figure 6 jcm-11-07205-f006:**
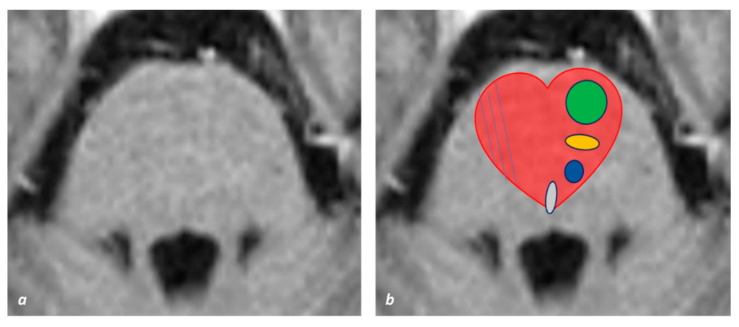
**Schematic view of pontine structures involved in the heart appearance sign. (a) Lower pontine axial section (T1W MRI). (b) Corresponding redesigned picture in (b) with the superimposed red heart sign. Arterial supply is depicted in red for paramedian branches, red with blue lines for anterolateral branches and white for long circumferential arteries. On the right half of the heart are depicted the corticospinal tract (green), the medial lemniscus (yellow), the central tegmental bundle (blue) and the medial longitudinal fasciculus (violet)**.

**Figure 7 jcm-11-07205-f007:**
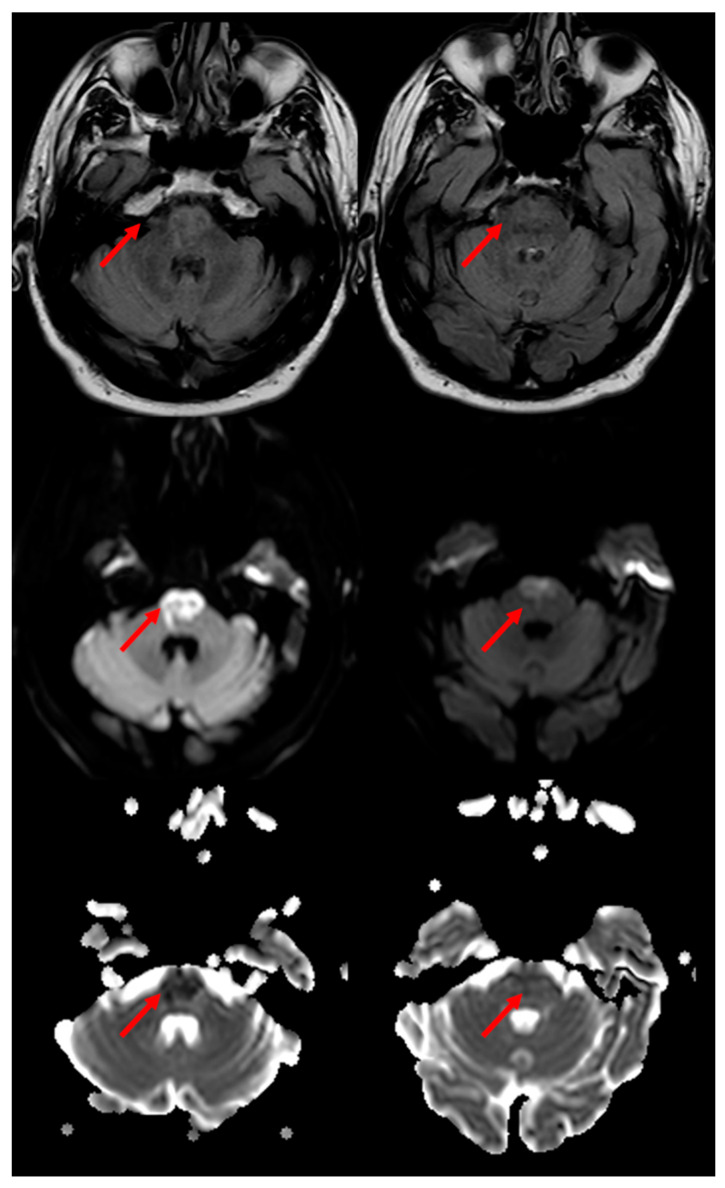
**“Heart appearance sign” in acute bilateral pontine paramedian infarction, which is well evident in MRI on FLAIR axial images (first row), DWI images (second row) and Apparent Diffusion Coefficient (ADC) images (third row) (red arrows)**.

**Table 1 jcm-11-07205-t001:** **Arterial territories of the medulla and main arterial supply**.

Arterial Territories	Main Supplying Arteries
**Anteromedial territory**	It is supplied by the ASA and VA cranially, and the ASA, inferiorly.
**Anterolateral territory**	It is supplied by the ASA and PICA inferiorly, and by the ASA and VA, superiorly.
**Lateral territory**	It is supplied by the PICA, BA, VA and AICA from the caudal to cranial regions.
**Posterior territory**	It is supplied by the PSAs inferiorly and the PICA, superiorly.

Abbreviations: basilar artery (BA), vertebral artery (VA), anterior spinal artery (ASA), posterior spinal artery (PSA), posterior inferior cerebellar artery (PICA) and anterior inferior cerebellar artery (AICA).

**Table 2 jcm-11-07205-t002:** **Arterial territories of the pons and main arterial supply**.

Arterial Territories	Main Supplying Arteries
**Anteromedial territory**	It is supplied by the pontine perforating arteries arising from the BA and entering from the basilar sulcus.
**Anterolateral territory**	It is supplied by the perforating branches arising from the AICA and entering at the pontomedullary sulcus.
**Lateral territory**	It is supplied by lateral pontine perforators that arise directly from the BA, AICA or SCA.

Abbreviations: basilar artery (BA), anterior inferior cerebellar artery (AICA) and superior cerebellar artery (SCA).

**Table 3 jcm-11-07205-t003:** **Anatomical portions of the medulla oblongata**.

Portions	Structures
**Anterior portion**	Fibers of the corticospinal tract, which most of them cross over to the contralateral side.
**Tegmentum**	Olivary complex, nuclei of cranial nerves (VIII–XII and part of V), parts of the reticular formation and ascending and descending fiber tracts (e.g., sympathetic fibers).
**Posterior portion**	The lower part is anatomically similar to the spinal cord, and it contains ascending fiber tracts that mostly end in the nuclei gracilis and cuneatus.

**Table 4 jcm-11-07205-t004:** **Structures involved in MMI and corresponding neurological deficit (modified from [10])**.

Structures	Main Neurological Deficit
**Corticobulbar tract**	Dysarthria.
**Corticospinal tract**	Contralateral hemiparesis/hemiplegia.
**Medial Lemniscus**	Contralateral loss of position sense and vibration.
**Medial longitudinal fasciculus (MLF)**	Ipsilateral internuclear ophtalmoplegia.
**Hypoglossal (CN XII) nucleus**	Ipsilateral weakness of the hemitongue.

Abbreviations: Medial Medullar Infarction (MMI).

**Table 5 jcm-11-07205-t005:** **Structures involved in anteromedial pontine infarction and corresponding neurological deficit (modified from [10])**.

Structures	Main Neurological Deficit
**Corticobulbar tract**	Dysarthria, contralateral facial palsy.
**Corticospinal tract**	Contralateral hemiparesis/hemiplegia.
**Corticopontine tract**	Contralateral ataxia, pathologic laughter.
**Medial Lemniscus**	Contralateral loss of position sense and vibration.
**Medial longitudinal fasciculus (MLF)**	Ipsilateral internuclear ophtalmoplegia.
**CN VI nucleus, CN VII fibers**	Ipsilateral lateral rectus palsy, ipsilateral facial weakness.
**Paramedian pontine reticular formation (PPRF)**	Ipsilateral horizontal gaze paresis.

**Table 6 jcm-11-07205-t006:** Perforating branches of the interpeduncular fossa.

Branches	Supplying Area and Associated Clinical Syndrome
**PThA**	Paramedian thalamus. Thalamo-peduncular syndrome or Percheron’s artery syndrome.
**SPMA**	Rostral midbrain (paramedian region). Thalamo-peduncular syndrome or Percheron’s artery syndrome.
**IPMA**	Caudal midbrain (paramedian region). Wernekink’s commissure syndrome.

**Abbreviations:** Paramedian thalamic arteries (PThA); Superior paramedian mesencephalic arteries (SPMA); Inferior paramedian mesencephalic arteries (IPMA).

**Table 7 jcm-11-07205-t007:** Clinical hints for cerebrovascular vs. spinal cause of acute onset quadriplegia.

Issues	Main Interpretations
**Onset of symptoms**	An acute or hyperacute onset with a very short time to the maximal deficit is more frequent in brain than it is in spinal vascular disorders.
**Previous clinical history**	A previous transient neurological symptom might suggest the site of brain disfunction and it is more frequent in brain then in spinal vascular disorders.
**Facial sparing**	It may occur in both brain and spinal vascular disorders.
**Additional neurological signs**	In brainstem involvement, oculomotor or cerebellar signs may be present and help to localize the site of the lesion.
**Extension of deficits after the onset of symptoms**	In brainstem vascular lesions is not uncommon the sequential occurrence of additional deficits due to the progressive extension of ischemia to the entire territory of the affected artery.

## Data Availability

Not applicable.

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
