# Peer review of "Acute Onset Quadriplegia and Stroke: Look at the Brainstem, Look at the Midline"

_jcm, 2022, doi:10.3390/jcm11237205_

Round 1
Reviewer 1 Report
Thank you for preparing this topic. A few more examples to do justice to the diversity of causes and morphology in these cases would be desirable. In addition, a table (as conclusion) that again clearly shows (e.g. in bullet points/flow chart) how an optimal management of this patient group could look like.
Author Response
We would thank the reviewer for reading our paper and for the suggestions.
Thank you for preparing this topic. A few more examples to do justice to the diversity of causes and morphology in these cases would be desirable.
We carefully considered the suggestion of the reviewer and we would like to thank him/her for it, but we think that it could take us outside the main aim of the paper and make the paper itself too long and less useful for a general reader. The aim of our paper was to integrate in a pragmatical manner the clinical, neuroanatomical and neuroimaging features of ischemic stroke locations associated with quadriplegia adding two clinical examples of the two infarct locations associated with this clinical presentation. The main focus is to raise attention to the diagnosis and not to discuss the general management of these rare patients or the cause (atherosclerosis vs cardioembolic vs SVD or other causes). These are the main reasons why we decided to provide a focused neuroanatomical view mainly on the vascular side in order to help the clinicians to understand when to think on brain and not spine (or peripheral nervous system) facing with an acute onset quadriplegia. As supported by the presented literature, this clinical manifestation and the corresponding infarct locations is extremely rare and less known but it should not be missed. We detailed in the text, figures and tables the main brainstem structures involved in the corresponding lesions, their vascular anatomy with its extreme variability and the derived neurological signs and presentations. We did not present an example of midbrain midline infarction because it is not associated to quadriplegia/quadriparesis. The rarity of quadriplegia as presenting sign of brainstem stroke makes very hard to see it in clinical practice.
In addition, a table (as conclusion) that again clearly shows (e.g. in bullet points/flow chart) how an optimal management of this patient group could look like.
We thank the reviewer for his/her suggestion. We added in a final table the main clinical issues for the differential diagnosis as in the corresponding paragraph in order to summarize some practical conclusions.
Reviewer 2 Report
Dear authors,
I have the following suggestions:
- line 34 - remove ''to'' from ''might help to recognize''
- lines 45-46 - the sentence is ambiguous and should be rephrased
- figure 4 - arrows pointing the ''heart appearance sign'' would be useful
- lines 248 -253 - the paragraph seems discordant to the rest of the text
- figure 5 and figure 7 - I suggest to insert arrows pointing the changes
- please discuss about the therapeutic decisions in the particular clinical entities that you presented
Author Response
First of all, we would like to thank the reviewer fro the appreciaton of our paper and for the suggestions.
- line 34 - remove ''to'' from ''might help to recognize'' (DONE)
- lines 45-46 - the sentence is ambiguous and should be rephrased (DONE)
The new sentence is: "Quadriplegia has been described in spinal cord injuries across a wide range of causes, including spinal cord ischemia"
- figure 4 - arrows pointing the ''heart appearance sign'' would be useful (DONE and the caption has been changed accordingly)
- lines 248 -253 - the paragraph seems discordant to the rest of the text
We changed the incipit of the sentence ("The patient whose MRI was shown in Figure 4 ...") in order to integrate the description in the paragraph.
- figure 5 and figure 7 - I suggest to insert arrows pointing the changes (DONE and the captions were updated accordingly)
- please discuss about the therapeutic decisions in the particular clinical entities that you presented
We added the detail for the first case: "
The treatment strategy in the acute phase was a double antiplatelets regimen with high dosage statin associated to low-molecular weight heparin (LMWH) for the prevention of venous thromboembolism."
The second case was similar becaus of the extensive atherothrombotic disease.
Thanks again for your appreciation.
Reviewer 3 Report
This manuscript provides detailed descriptions of artery territories associated with various symptoms as well as projections of corticospinal tracts in different parts of the brain. The figures and tables are self-explanatory. The authors have also provided further information in symptoms in addition to the visible signs in the MRI (e.g., the Mickey Mouse ears sign). The contribution of this study is that it suggests the problem in the brainstem and midline in the rare case of acute atraumatic quadriplegia in the absence of spinal cord lesions. This draws attention to the aspects that can help in selecting the correct diagnostic course.
In the section of clinical hints an differential diagnosis, the authors listed the five aspects to be considered for diagnostic directions. These are very helpful. I would suggest that adding a table to summarize this section. For example, the primary neurological deficit and the constant element can be noted in the table.
Author Response
First of all, we would like to thank the reviewer for the appreciation of our paper.
We add a further table (table 7) in the paragraph on clinical hints.
Table 7. Clinical hints for cerebrovascular vs spinal cause of acute onset quadriplegia.
Issues |
Main interpretations |
Onset of symptoms |
An acute or hyperacute onset with a very short time to the maximal deficit is more frequent in brain than in spinal vascular disorders |
Previous clinical history |
A previous transient neurological symptom might suggest the site of brain disfunction and it is more frequent in brain then in spinal vascular disorders |
Facial sparing |
It may occur in both brain and spinal vascular disorders |
Additional neurological signs |
In brainstem involvement oculomotor or cerebellar signs may be present and help to localize the site of the lesion |
Extension of deficits after the onset of symptoms |
In brainstem vascular lesions is not uncommon the sequential occurrence of additional deficits due to the progressive extension of ischemia to the entire territory of the affected artery |